# Rare Coding Variants in Patients with Non-Syndromic Vestibular Dysfunction

**DOI:** 10.3390/genes14040831

**Published:** 2023-03-30

**Authors:** Angelo Augusto M. Sumalde, Melissa A. Scholes, Olivia A. Kalmanson, Elizabeth A. Terhune, Lidia Frejo, Cambria I. Wethey, Pablo Roman-Naranjo, Patrick M. Carry, Samuel P. Gubbels, Jose A. Lopez-Escamez, Nancy Hadley-Miller, Regie Lyn P. Santos-Cortez

**Affiliations:** 1Department of Otolaryngology-Head and Neck Surgery, School of Medicine, University of Colorado Anschutz Medical Campus, Aurora, CO 80045, USA; 2Department of Otolaryngology-Head and Neck Surgery, University of the Philippines Manila College of Medicine, Philippine General Hospital, Manila 1000, Philippines; 3Department of Pediatric Otolaryngology, Children’s Hospital Colorado, Aurora, CO 80045, USA; 4Department of Otolaryngology-Head and Neck Surgery, University of Mississippi Medical Center, Jackson, MS 39216, USA; 5Department of Orthopedics, School of Medicine, University of Colorado Anschutz Medical Campus, Aurora, CO 80045, USA; 6Otology and Neurotology Group CTS495, Department of Genomic Medicine, GENYO-Centre for Genomics and Oncological Research-Pfizer-University of Granada-Junta de Andalucia, PTS, 18016 Granada, Spain; 7Division of Otolaryngology, Department of Surgery, Instituto de Investigación Biosanitaria, ibs.GRANADA, Universidad de Granada, 18071 Granada, Spain; 8Musculoskeletal Research Center, Children’s Hospital Colorado, Aurora, CO 80045, USA; 9Meniere’s Disease Neuroscience Research Program, Faculty of Medicine & Health, School of Medical Sciences, The Kolling Institute, University of Sydney, Sydney, NSW 2006, Australia; 10Center for Children’s Surgery, Children’s Hospital Colorado, Aurora, CO 80045, USA

**Keywords:** balance disorder, dizziness, *HMX3*, *LAMA2*, Meniere’s disease, *OTOP1*, rare variant, semicircular canal, vertigo, vestibular

## Abstract

Vertigo due to vestibular dysfunction is rare in children. The elucidation of its etiology will improve clinical management and the quality of life of patients. Genes for vestibular dysfunction were previously identified in patients with both hearing loss and vertigo. This study aimed to identify rare, coding variants in children with peripheral vertigo but no hearing loss, and in patients with potentially overlapping phenotypes, namely, Meniere’s disease or idiopathic scoliosis. Rare variants were selected from the exome sequence data of 5 American children with vertigo, 226 Spanish patients with Meniere’s disease, and 38 European–American probands with scoliosis. In children with vertigo, 17 variants were found in 15 genes involved in migraine, musculoskeletal phenotypes, and vestibular development. Three genes, *OTOP1*, *HMX3*, and *LAMA2*, have knockout mouse models for vestibular dysfunction. Moreover, *HMX3* and *LAMA2* were expressed in human vestibular tissues. Rare variants within *ECM1*, *OTOP1*, and *OTOP2* were each identified in three adult patients with Meniere’s disease. Additionally, an *OTOP1* variant was identified in 11 adolescents with lateral semicircular canal asymmetry, 10 of whom have scoliosis. We hypothesize that peripheral vestibular dysfunction in children may be due to multiple rare variants within genes that are involved in the inner ear structure, migraine, and musculoskeletal disease.

## 1. Introduction

Dizziness is a common symptom encountered in clinical practice, affecting approximately 20–30% of the population [1]. Illusionary states of dizziness or vertigo point to disorders of the vestibular system in the inner ear, and vestibular disorders are defined as a group of diseases leading to the transient or permanent loss of the vestibular function [2]. Pediatric vestibular diseases can be debilitating and inhibit children from participation in physical activities as well as affect their performance in school. The diagnosis and management of these diseases pose a challenge to clinicians due to variations in clinical presentation and overlaps in symptomatology among different disease states. Additionally, the interaction of the vestibular system with other organs of balance such as the neck and spine can further add to the complexity of managing vestibulopathy. For example, recent evidence from animal models and human clinical studies supported an association between vestibular disease and adolescent idiopathic scoliosis (AIS), such that the unilateral lesions of the otolithic and semicircular canal organs resulted in vertebral rotation and head tilt [3].

Despite the evidence of familial clustering in vestibular diseases, including Meniere’s disease [4] and vestibular migraine [5], studies on the genetic basis of vestibular diseases are few compared to those on sensorineural hearing loss. Genetic studies on possible candidate genes in vestibular diseases often involve those associated with syndromic and non-syndromic sensorineural hearing loss [2], with vestibular conditions initially not being considered as the primary symptoms. Because of the paucity of knowledge on the etiology of vestibular disorders, particularly in children, diagnoses are more frequently attributed to migraine rather than innate vestibulopathy [6]. Given the complex presentations of vertigo, we hypothesized that a combination of rare variants in a few genes, including candidate genes for inner ear disorders and migraine, may be present in each pediatric patient and contribute to the rarity and earlier onset of vestibular dysfunction in children, similar to what is observed in other pediatric immune or neurologic diseases [7,8]. With the availability of high-throughput massively parallel sequencing technology, we sought to identify genetic variants in pediatric patients with vestibular and balance disorders but not hearing loss or other syndromes.

To this end, this study aimed to identify rare, potentially deleterious coding variants using exome sequence data from patients with early onset non-syndromic vestibular dysfunction. From the exome sequence data, we identified rare variants in the genes involved in migraine, inner ear, and musculoskeletal conditions in American children with peripheral vestibulopathy and performed a network analysis for the identified genes. Due to the rarity of non-syndromic peripheral vertigo in children, we screened for rare or low-frequency variants in the same candidate genes using sequence data from two cohorts with potentially overlapping phenotypes: (1) familial and sporadic Meniere’s disease patients from Spain; and (2) European–American families with AIS, including children with asymmetry of the lateral semicircular canal of the inner ear as documented by MRI. Finally, we demonstrated the expression of two genes, namely, *HMX3* and *LAMA2*, in human vestibular tissues.

## 2. Materials and Methods

### 2.1. Ethical Approvals and Subject Recruitment

All studies were performed according to the ethical principles as stated in the Declaration of Helsinki. Prior to the study’s initiation, this study was approved by the Colorado Multiple Institutional Review Board (protocols 06-1161, 07-0417, 09-0706, 17-1720, 19-1930, and 21-3246) and the Comite de Etica de Investigacion Clinica de Granada H2020-SC1-2019-848261. Informed consent was obtained from each family member who provided DNA samples and clinical data, including informed consent from the parents of children who participated in the study, and from each adult patient who provided vestibular tissue samples. Child assent was also documented as applicable.

Five patient cohorts were included in this study (Appendix A):Cohort 1: For the pilot study, five children who were diagnosed with vestibular dysfunction and had a family history of vertigo were recruited into the study (Table 1). Patients were excluded if there was a diagnosis of hearing loss, known neurologic disorders, or other significant comorbidities. The five enrolled patients did not have any history of head trauma, notable past medical, prenatal, and neonatal events, and inner ear abnormalities seen by 2D high-resolution temporal bone CT or MRI. The results of audiometric testing and tympanometry were within the normal limits. Saliva samples were obtained from the study participants using Oragene saliva kits (DNA Genotek, Ottawa, ON, Canada). Genomic DNA was isolated from saliva using the manufacturer’s protocol.
genes-14-00831-t001_Table 1Table 1Pediatric patients with peripheral vestibular disorders.ID ^1^Age at Initial Consult SexEthnicityFamily History of MigraineDuration of SymptomsClinical Diagnosis16 yearsFWhite non-HispanicNo2 yearsBenign paroxysmal vertigo of childhood211 yearsMHispanicNo5 monthsVertigo313 yearsFWhite non-HispanicYes9 yearsVestibular migraine49 yearsFWhite non-HispanicYes6 yearsVestibular migraine5 ^2^12 yearsFWhite non-HispanicYes8 yearsLeft-sided vestibular weakness^1^ All five patients did not suffer from hearing loss, or family history of hearing loss or vertigo. ^2^ Patient 5 also has vitiligo. The four other patients did not have any known comorbidities.
Cohort 2: Meniere’s disease patients were recruited and diagnosed from different Spanish hospitals following the diagnostic criteria defined by the International Classification Committee for Vestibular Disorders of the Barany Society [9]. Other vestibular diseases were excluded after performing a complete audio-vestibular assessment, including MRI. The DNA from these patients were collected and extracted from blood or saliva using the QIAamp DNA Mini Kit (Qiagen, Venlo, the Netherlands) and Oragene saliva kits, respectively.Cohort 3: European–American families with AIS were enrolled as previously described [10,11,12]. The families were selected for sequencing if the proband had no co-occurring congenital deformity or genetic disorder and was diagnosed with AIS based on a standing anteroposterior spinal radiograph showing ≥10° curvature by the Cobb method with pedicle rotation. In total, 3 to 4 AIS-affected individuals from each of the 28 families were selected for exome sequencing based on the scoliotic curve severity, the availability of DNA samples, and the genetic distance across the individuals [10,11].Cohort 4: A second cohort of individuals with AIS (Appendix A) were enrolled as above with added consent to obtain the morphological measurements of the semicircular canals using the three-dimensional reconstruction of images from T2-weighted MRI [13]. This previous study was performed to examine the association between the occurrence of semicircular canal defects and spine deformity. From cohort 4, eleven individuals with differences in angulation between the axial plane and the lateral semicircular canal when comparing the right and left inner ears (from here on referred to as LSCC asymmetry) were selected for Sanger sequencing of exons within two genes: *HMX3* and *OTOP1*. Of note, out of eleven selected individuals with subtle changes in their inner ear structure, ten were previously diagnosed with AIS (Appendix A), while the eleventh individual was designated as an AIS control due to normal spine curvature.Cohort 5: Two American patients with vestibular schwannomas who underwent tumor resection via a translabyrinthine approach provided vestibular tissue samples that would otherwise be discarded during surgery. One patient self-identified as White/Caucasian, while the second patient refused to disclose their ethnicity. Both patients did not experience any preoperative symptoms of vertigo, suggesting normal vestibular function despite the ipsilateral nerve sheath tumor. Immediately upon removal, the vestibular tissue samples were placed in RNAlater (Invitrogen/Thermo Fisher, Waltham, MA, USA). The samples were stored at 4 °C and processed within 30 days.

### 2.2. Exome and Sanger Sequencing of DNA Samples

Exome sequencing was performed using DNA samples from the cohorts 1 to 3. The DNA samples from five American children with vestibular disorders {cohort 1} were submitted to the Northwest Genomics Center, University of Washington, for exome sequencing. The sequence capture was performed using the Twist Bioscience Human Core Exome Kit (South San Francisco, CA, USA) and an Illumina NovaSeq (San Diego, CA, USA). DNA samples from the Meniere’s disease cohort {cohort 2} were submitted for exome sequencing at 100× coverage using the Agilent SureSelect XT v6 Exome kit (Agilent Technologies, Santa Clara, CA, USA) and the Illumina NovaSeq 6000 platform. For the European–American family members with AIS {cohort 3}, DNA was extracted from the blood samples. Exome sequencing was completed using the Agilent SureSelect Human V5 (51 Mb) exon capture kit on an Illumina HiSeq 2500 at the Otogenetics Corporation (Atlanta, GA, USA), with a minimum average coverage of 50×. For all three cohorts, Burrows–Wheeler Aligner [14] was used to align the generated sequence data to the human reference sequence (hg19 for the US children with vestibular disease and the Spanish Meniere’s disease cohort, hg38 for the AIS cohort) and produce the .bam files. Variant calling was performed and the .vcf files were created using the Genome Analysis Toolkit [15].

For the validation of the candidate variants, Sanger sequencing was performed for rare variants that were selected from exome sequence data using the same five DNA samples from American children with vestibular dysfunction and available maternal DNA samples (Table 2). The list of genes that harbored these variants were then used to select rare predicted-to-be-damaging variants from the exome sequence data of individuals with Meniere’s disease or AIS. For the Meniere’s disease cohort, the variants identified in the probands were Sanger-sequenced in the rest of the DNA samples that were available from family members. In addition, exon 2 of *HMX3* (NM_001105574.2) and exon 6 of *OTOP1* (NM_177998.3) were Sanger-sequenced using eleven DNA samples from cohort 4 including 11 European–American probands with LSCC asymmetry, ten of whom had AIS.

### 2.3. Variant Prioritization and Gene Network Analysis

Exome sequence data from the five American children with vestibular dysfunction {cohort 1} were annotated using ANNOVAR (annovar.openbioinformatics.org, accessed on 8 March 2023) [16]. Variants were selected if they lie within genes known for: (a) hearing loss in humans (i.e., 150 syndromic and non-syndromic genes in a targeted genetic screening panel Otoscope v8 plus newly published genes); (b) mouse models with vestibular and/or hearing defects; (c) high expression in mouse vestibule based on the gene Expression Analysis Resource database (umgear.org, accessed on 8 March 2023); and (d) association with migraine (Appendix A). The exome sequence data were also annotated using MutationTaster [17]; however, no potentially “disease-causing” indels were identified. In Colorado, the 10-year prevalence of pediatric vertigo without migraine (whether central or peripheral) was estimated at 2.5%, and for the rarer phenotype of co-occurring migraine and vertigo at 0.8% (unpublished data from the COMPASS database). Additionally, around half of the children who consulted at the same specialty clinic for vertigo were diagnosed with peripheral vestibular disease. Based on these estimates, single nucleotide variants were selected for further study if they have a: (a) minor allele frequency (MAF) < 0.001 in all populations in the Genome Aggregation Database (gnomAD v3.1.2, gnomad.broadinstitute.org, accessed on 8 March 2023), the Greater Middle East (GME) Variome database (igm.ecsd.edu/gme), Bravo (bravo.sph.umich.edu, TOPMed freeze 5), and All of Us (databrowser.researchallofus.org, accessed on 8 March 2023); (b) a scaled score ≥ 15 in the Combined Annotation-Dependent Depletion (CADD) database (cadd.gs.washington.edu, GRCh38-v1.6); and (c) prediction as damaging or deleterious by at least one bioinformatic tool in dbNSFP v4.1 (sites.google.com/site/jpopgen/dbNSFP). The bioinformatics tools in dbNSFP included FATHMM, fathmmMKL, a likelihood ratio test (LRT), M-CAP, metaLR, metaSVM, MutationTaster, PolyPhen2 HDIV/HVAR, PROVEAN, and SIFT (Table 2). Variants that satisfied these criteria were excluded if, based on the known literature from human and animal studies, the genes that harbored these variants were less likely to explain the patient’s presenting symptoms (Appendix A).
genes-14-00831-t002_Table 2Table 2Genetic variants identified in children with peripheral vestibular disorders.IDGeneRefSeq NM_#VariantHighest MAF in Databases ^1^Damaging PredictionScaled CADD ScoreKnown Literature on Gene [References]MIM#1*TMEM130*001134450c.52C>T (p.(Leu18Phe))0SI, MC24.1High expression in developing mouse utricular hair cells [18]n.a.1*LRP1* ^2^002332c.13471G>C (p.(Asp4491His))(gA)OTH: 9.5 × 10^–4^ppHD, ppHV, LRT, MT24.8Associated with migraine [19]1077701*ECM1* ^2^001202858c.1126T>C (p.(Cys376Arg))(AU)Lat:9.9 × 10^–5^SI, ppHD, ppHV, LRT, MT, FA, PR, mSVM, mLR, MC25.5Associated with migraine [20,21]6022011*CARF* ^2^001352679c.1718C>T (p.(Ser573Phe))(AU)OTH: 1.1 × 10^–4^SI, ppHD, ppHV, MT27.9Associated with migraine [20,22]6075862*CTIF*014772c.1547A>G (p.(Gln516Arg))(AU)Eur: 3.95 × 10^–4^LRT, fMKL20.3Associated with migraine [19]6131782*KIF6*001289024c.1838G>A (p.(Arg613Gln))(AU)Eur: 7.3 × 10^–5^ppHD, MT, fMKL23.6High expression in developing mouse utricular hair cells [18]6139192*KLC2* ^2^001134774c.976C>T (p.(Gln326*))0LRT, fMKL38.0Knockdown in zebrafish resulted in twisted tail and inability to swim [23]6117292*OTOP2*178160c.379A>G (p.(Ser127Gly))(gA)NFE: 1.5 × 10^–5^MT, fMKL22.0Maps to a candidate region of USH1G [24]6078273*HMX3*001105574c.534G>T (p.(Glu178Asp))Bravo: 1.6 × 10^–5^ppHD, LRT, MT, FA, mLR, MC17.8KO mice exhibit abnormal circling behavior and vestibular defects [25]6133803*LRP1*002332c.13169A>C (p.(Asp4390Thr))0fMKL22.0Associated withmigraine [19]1077704*MYO5A* ^2^000259c.3257G>A (p.(Ser1086Asn))0LRT, fMKL20.6Associated with Elejalde syndrome; involved in neurologic development [26]2769034*MYO5A* ^2^000259c.343C>A (p.(Gln115Lys))0SI, LRT, MT, FA, mLR, MC, fMKL23.54*RNF213*001256071c.9197C>T (p.(Pro3066Leu))(AU)Eur: 4.2 × 10^–5^SI, LRT, MT, PR, fMKL25.3Associated withmigraine [22]6137684*APOE*000041c.875G>T (p.(Arg292Leu))(AU)Eur: 2.1 × 10^–5^SI, ppHD, ppHV, PR, MC, fMKL31.0Associated withmigraine [27,28]1077415*LAMA2* ^2^000426c.6229G>A (p.(Ala2077Thr))(gA)NFE: 4.6 × 10^–4^MT, fMKL16.9Abnormal cochlear and vestibular morphology in homozygous *dy/dy* mice [29]1562255*SLC35D2*007001c.601A>G (p.(Lys201Glu))(gA)NFE: 1.5 × 10^–5^SI, LRT, MT, PR, MC, fMKL29.1Associated with migraine [19]6091825*OTOP1*177998c.164A>G (p.(Gln55Arg))(AU)Eur: 1.98 × 10^–4^MT, MC17.3Associated with non-syndromic vestibular disorder with otoconial agenesis in mice [30]607806^1^ Highest MAF in gnomAD, GME, Bravo, or All of Us. ^2^ Present in maternal DNA. Abbreviations: (AU): All of Us database; CADD: Combined Annotation-Dependent Depletion; Eur: European; FA: FATHMM; fMKL: fathmmMKL; (gA): gnomAD/Genome Aggregation Database; KO: knockout; Lat: Latin American; LRT: likelihood ratio test; MAF: minor allele frequency; MC: M-CAP; mLR: metaLR; mSVM: MetaSVM; MT: MutationTaster; n.a., not available; NFE: non-Finnish European; OTH: other; ppHD: PolyPhen2 HDIV; ppHV: PolyPhen2 HVAR; PR: PROVEAN; SI: SIFT.


The Variant Effect Predictor was used to annotate the Meniere’s disease dataset from cohort 2 [31]. Variants were selected if they were found within the genes identified in American children with vestibular disease, and if they are rare and predicted to be damaging based on the criteria outlined above.

For cohort 3 with AIS, variants were identified from the exome sequence data using FreeBayes then filtered using SnpEff version 4.1 [32]. Custom scripts were used to select non-synonymous, coding indel, and splice variants with MAF ≤ 0.05, which lie within the genes listed in Table 2. Single nucleotide variants were further selected if deemed damaging by at least one prediction tool in dbNSFP v3.0.

Network analysis was performed using NetworkAnalyst (networkanalyst.ca, accessed 25 March 2023) [33] and the InnateDB [34] database was used to identify protein–protein interactions for genes identified to have variants among the children with peripheral vestibular dysfunction. Enrichment analysis was performed using the PANTHER, Gene Ontology, and REACTOME databases within the NetworkAnalyst website, while noting significant pathways using the false discovery rate (FDR)-adjusted-*p* < 0.05.

### 2.4. RNA Isolation and RT-PCR for Human Vestibular Tissues

RNA was isolated from human vestibular tissues using the RNeasy Micro Kit (Qiagen, Hilden, Germany) according to the manufacturer’s protocol, with the addition of a pre-homogenization step using glass beads (2 mm and 3 mm) and the MO BIO Vortex Adapter (Carlsbad, CA, USA). The SuperScript IV First-Strand Synthesis System with ezDNase Enzyme (Thermo Fisher Scientific, Waltham, MA, USA) was used for cDNA synthesis. For two genes, *HMX3* and *LAMA2*, RT-PCR was performed using the DreamTaq Green PCR Master Mix (Thermo Fisher) with an annealing temperature of 60 °C and 35 cycles. PCR products were run on a 4% agarose gel and the Universal Human Reference RNA (Invitrogen/Thermo Fisher) was used as the positive control.

## 3. Results

### 3.1. Rare Genetic Variants in Pediatric Patients with Early Onset Vestibular Dysfunction

Variants were initially identified from the exome sequence data of five children with vestibular dysfunction (cohort 1, Table 1). Seventeen rare coding heterozygous variants, predicted to be deleterious, were identified (Table 2). All fifteen genes have a moderate-to-high expression in developing mouse vestibules (umgear.org, accessed on 8 March 2023). Seven out of the seventeen rare variants were found in genes which were previously associated with migraine in genome-wide or candidate gene association studies [19,20,22,27,28] that used common variant data from adult patients (Table 2). A description of each patient’s clinical profile in light of the identified rare variants is hereby presented.
Patient 1: For two years prior to consult, a six-year-old female experienced intermittent vertigo lasting 15–60 min, occurring 2–3 times per month, and without associated tinnitus, hearing loss, headaches, and visual changes. She had abnormal bilateral findings with the head impulse test and was diagnosed with benign paroxysmal vertigo of childhood (BPVC). Exome sequencing revealed that she had three rare variants in migraine genes, *LRP1* c.13471G>C (p.(Asp4491His)), *ECM1* c.1126T>C (p.(Cys376Arg)), and *CARF* c.1718C>T (p.(Ser573Phe)) [19,20,21,22]. At the time of consult, patient 1 did not exhibit migraine symptoms; however, BPVC is considered a migraine precursor [35]. An additional variant in a gene highly expressed in developing mouse utricular hair cell [18], *TMEM130* c.52C>T (p.(Leu18Phe)), was also found.Patient 2: An 11-year-old male reported multiple episodes of vertigo lasting minutes to hours during the last five months prior to consultation. He did not have tinnitus, hearing loss, or headaches. His head impulse test had abnormal findings bilaterally, and caloric testing revealed a 21% reduced response on the right side. Rare variants were noted in genes previously associated with neurodevelopmental disorders [23,36]: *KIF6* c.1838G>A (p.(Arg613Gln)) [36] and *KLC2* c.976C>T (p.(Gln326*)). In addition, he had a rare variant in another migraine-associated gene, *CTIF* c.1547A>G (p.(Gln516Arg)) [19]. A fourth rare variant, c.379A>G (p.(Ser127Gly)), was found within *OTOP2*, a gene which maps to the candidate region of Usher syndrome 1G (USH1G) [24], which includes congenital hearing loss, vestibular areflexia, and adolescent-onset retinitis pigmentosa. Note that patient 2 does not have Usher syndrome, hearing loss, or eye disease.Patient 3: A 13-year-old female presented with on-and-off vertigo in the previous nine years and episodes of migraine headaches separate from the vertigo attacks. The clinical evaluation revealed an abnormal right-sided head impulse test. Based on her presentation, she was diagnosed with probable vestibular migraine. A damaging variant in the homeobox family protein *HMX3* c.534G>T (p.(Glu178Asp)) was identified from her exome sequence data. The knockout of *HMX3* in mice led to abnormal circling behavior and severe structural defects in the inner ear vestibule [25]; however, prior to this work, there was no strong evidence of vertigo-causal *HMX3* variants in humans [37]. In addition, a rare variant, c.13169A>C (p.(Asn4390Thr)), was identified within the migraine-associated gene *LRP1* and was deemed pathogenic by the bioinformatic predictor fathmmMKL.Patient 4: A nine-year-old female had recurrent, severe headaches which were unresponsive to a drug cocktail containing Naproxen, Prochlorperazine, and Diphenhydramine. Her headaches were associated with photophobia, phonophobia, vertigo, nausea, vomiting, and occasional tinnitus. The clinical evaluation revealed bilateral torsional and rotational gaze nystagmus, and an abnormal head impulse test on the right. Based on her presentation, she was diagnosed with vestibular migraine. She has novel rare variants in two migraine-associated genes [22,27,28,38]: *APOE* c.875G>T (p.(Arg292Leu)) [26,33] and *RNF213* c.9197C>T (p.(Pro3066Leu)). Additionally, two variants within *MYO5A* c.3257G>A (p.(Ser1086Asn)) and c.343C>A (p.(Gln115Lys)), a gene previously implicated in neurodevelopment [26], were identified from her exome sequence data. The Sanger sequencing of maternal DNA revealed that these two *MYO5A* variants were inherited in cis.Patient 5: A 12-year-old female presented with an 8-year history of episodic vertigo, lasting minutes to hours, and occasionally occurring when performing gymnastics. The clinical evaluation revealed a bilaterally abnormal head impulse test and 50% left canal paresis on caloric testing. She was diagnosed with left-sided vestibular weakness. A rare missense variant c.6229G>A (p.(Ala2077Thr)) was identified in *LAMA2*, which is known for merosin-deficient congenital muscular dystrophy 1A (MDC1A; MIM 607855) [29]. Furthermore, *Lama2*-knockout mice demonstrated cochlear and vestibular malformations [29]. Additionally, two rare variants—c.601A>G (p.(Lys201Glu)) in the migraine-associated gene *SLC35D2* [19] and c.164A>G (p.(Gln55Arg)) within *OTOP1*—were identified in the exome data of patient 5. The knockout of *Otop1* previously resulted in non-syndromic vestibular defects and otoconial agenesis in mice [30].

### 3.2. Network Analysis

Network analysis was performed using as an input the 15 genes which harbored rare variants. The resulting subnetwork (Figure 1) included 9 genes: migraine-associated genes *APOE*, *LRP1*, *CARF*, *CTIF*, *RNF213*, and *ECM1*; *LAMA2*, which is associated with vestibular anomalies in mice; and the neurodevelopment-associated genes *MYO5A* and *KLC2*. Function analysis of this subnetwork revealed 23 significant PANTHER BP pathways, 120 GO BP pathways, and 98 REACTOME pathways. The top pathways from each analysis are listed in Table 3. The pathways common among these databases involved: membrane transport (i.e., membrane trafficking, vesicle mediated transport, exocytosis, receptor mediated endocytosis); the regulation of apoptosis (i.e., activation of BH3-only proteins, intrinsic pathway for apoptosis, activation of BAD to mitochondria, negative regulation of apoptotic processes); coagulation or hemostasis (i.e., hemostasis, blood coagulation, platelet activation, signaling and aggregation, platelet degranulation); and protein localization (i.e., translocation of GLUT4 to plasma membrane, protein localization, protein targeting). The pathways involved with membrane transport included four genes in the subnetwork, i.e., *APOE*, *LRP1*, *KLC2*, and *MYO5A*.

### 3.3. Rare Genetic Variants in Spanish Patients with Meniere’s Disease

A total of 226 patients with familial or sporadic Meniere’s disease {cohort 2} were screened for novel or rare variants in the 15 candidate genes identified in children with vestibular dysfunction (Table 2 and Table 4). Three genes, *ECM1*, *OTOP1*, and *OTOP2*, harbored rare variants in Spanish patients with Meniere’s disease. A rare *ECM1* variant c.844C>T (p.(Arg282Trp)) was identified in patient SANT19 who had familial Meniere’s disease but no migraine symptoms at the time of diagnosis. This variant was likewise identified in her father, who also suffered from Meniere’s disease. Additionally, a rare variant, *OTOP1* c.380A>T (p.(His127Leu)), was found in a sporadic patient suffering from bilateral Meniere’s disease since the age of 42. Within *OTOP2*, the c.760G>A (p.(Ala254Thr)) variant was identified in a sporadic patient with Meniere’s disease since early adulthood. Overall, these findings in the adult patients with Meniere’s disease in cohort 2 demonstrate that rare coding variants were identified in the same three genes as in the children with vertigo from cohort 1.

### 3.4. Genetic Variants in European–American Probands with AIS and LSCC Asymmetry

In the AIS cohorts {cohorts 3–4}, no variants in *HMX3* were identified in exome sequence data from cohort 3, which includes 28 families with AIS, or by Sanger sequencing of *HMX3* exon 2 in cohort 4 that has 11 children with LSCC asymmetry. From cohort 3, 3-4 AIS-affected individuals from each of the 2 families with ID 8 and 81 had the same variant *OTOP1* c.310C>A (p.(Leu104Met)). This variant had a MAF of 5% in GME Israel and 1% in gnomAD Finnish, a scaled CADD score of 23.9, and was predicted to be damaging by PolyPhen2 HDIV, MutationTaster, M-CAP, and fathmm-MKL_coding. While family 81 has isolated AIS with varying curvatures of the spine, family 8 has a known pathogenic *FBN1* variant c.7754T>C (p.Ile2585Thr) co-segregating with the AIS phenotype as well as cardiac phenotypes within the family. This latter finding makes a potential association between the *OTOP1* c.310C>A variant and AIS less likely.

On the other hand, a rare deleterious *OTOP1* c.29C>G (p.(Ser10Trp)) variant was identified in 11 adolescent individuals (cohort 4; Appendix A), all of whom had LSCC asymmetry by MRI, but only 10 of whom had AIS [13]. This rare *OTOP1* c.29C>G variant has a scaled CADD score of 14.98 and was deemed deleterious by SIFT. The identification of the *OTOP1* c.29C>G variant in 11 individuals with LSCC asymmetry, 1 of whom did not have AIS, strengthens a potential association between *OTOP1* and subtle vestibular defects.

### 3.5. HMX3 and LAMA2 Expression in Human Vestibular Tissues

Vestibular tissues, including the normal utricle, saccule, and semicircular canal ampullae from two patients with left-sided vestibular schwannoma {cohort 5}, were collected for the gene expression analysis. The first sample was taken from a 63-year-old female with ipsilateral moderate-to-profound sensorineural hearing loss and a feeling of disequilibrium without vertigo. The second sample was taken from a 60-year-old female with moderate-to-severe ipsilateral hearing loss without dizziness, imbalance, or vertigo. RT-PCR revealed that HMX3 and LAMA2 were strongly expressed in the collected human vestibular tissues (Figure 2).

In the gEAR database, 12 out of 15 genes in Table 2, including *HMX3* and *LAMA2*, were detected in bulk RNA-seq data from adult human vestibular sensory epithelia (Forge 2020 dataset). On the other hand, three genes, *CTIF*, *OTOP1*, and *OTOP2*, were not found in this human dataset from gEAR. It should be noted that in single-cell RNA-seq data from mice, homologs for all 15 genes were detected in: macrophages, hair cells, and neuronal cells of the crista ampullaris (Bermingham–Donough 2021 dataset); and Atoh1-induced utricular supporting cells (Groves 2019 dataset). In particular, *Otop1* was previously known to be expressed in macular epithelial cells and the overlying gelatinous membrane within the mouse vestibule [30].

## 4. Discussion

In total, 17 rare variants within 15 novel candidate genes were identified in 5 children with peripheral vestibular dysfunction (Table 1 and Table 2). Multiple rare variants were noted per child, with varied clinical presentations for each patient. These findings suggest that non-syndromic vertigo may be due to multiple rare variants contributing to early onset vertigo, which is rare in children. Of the candidate genes, rare variants were identified in *HMX3*, *LAMA2*, and *OTOP1*, which were shown in mouse models to be strong candidate genes for vestibular dysfunction. Additionally, we demonstrated that *HMX3* and *LAMA2* are highly expressed in human vestibular tissues (Figure 2). Rare variants in *OTOP1*, *OTOP2*, and *ECM1* were also identified in Spanish probands and families with adult-onset Meniere’s disease, although in each of these Spanish families, only one variant from our candidate genes was identified (Table 4). Furthermore, a network analysis revealed that migraine genes *CTIF*, *APOE*, *LRP1*, *RNF213*, *CARF*, and *ECM1*, musculoskeletal genes *KLC2* and *MYO5A*, as well as *LAMA2* were connected to a gene subnetwork, and some of these genes play a role in significant pathways for membrane transport, protein targeting, and coagulation (Figure 1, Table 3). Taken together, our findings support the hypothesis that multiple rare variants in the genes involved in the inner ear structure, migraine, and musculoskeletal development contribute to the early development of peripheral balance disorders.

Among the notable candidate genes are *OTOP1*, *HMX3*, and *LAMA2*. These genes were previously shown to cause vestibular dysfunction as observable behavioral changes as well as structural abnormalities in the inner ears of knockout mice [25,29,30]. *OTOP1* encodes an ion channel that is involved in maintaining the ionic environment during otoconia formation [30,39]. The mouse mutants *tilted* (*tlt*) and *mergulhador* (*mlh*) harbor single missense variants p.Ala151Glu and p.Leu408Gln within *Otop1* that led to an impaired balance and otoconial agenesis [30]. Histologically, the total absence of otoconia in the utricle or saccule or, more occasionally, few giant otoconia in the saccule were observed in *Otop1*-mutant mice [30]. However, the involvement of these genes in human vestibular dysfunction is less studied. Recently a common variant rs2272744 upstream of *OTOP1* was associated with vertigo in adult individuals of European descent in a genome-wide meta-analysis [40]. In our cohort 1, patient 5, who has a rare *OTOP1* missense variant (Table 2), reported vertigo symptoms during gymnastic maneuvers, suggesting possible otolith dysfunction in the semicircular canals. Additionally, a male Spanish patient with the onset of symptoms of Meniere’s disease at 42 years old and a maternal family history of vertigo and deafness also had a rare *OTOP1* missense variant (Table 4). Last, an *OTOP1* c.29C>G (p.(Ser10Trp)) variant was identified in 11 adolescents with LSCC asymmetry (Appendix A), only 10 of whom had AIS [13]. AIS is a common adolescent spinal deformity of unknown etiology; however, various studies have suggested abnormal somatosensory and vestibular integration as possible mechanisms that result in altered posture and balance [41,42]. Overall, these findings indicate that *OTOP1* is a strong candidate gene for non-syndromic vestibular dysfunction in humans.

Two other genes, *HMX3* and *LAMA2*, were identified to have rare missense variants in children with vertigo and are strongly expressed in human vestibular tissues (Figure 2). Additionally, mouse models strongly suggest these as candidate genes for vestibular dysfunction. *Hmx3*-knockout mice demonstrated vestibular defects as abnormal circling behavior and structural malformations in the form of utricle and saccule fusion with a significant reduction in sensory epithelia and the absence of the horizontal semicircular canal crista [25]. Although less well-described, human studies also suggest *HMX3* involvement in vertigo. A hemizygous deletion in chromosome 10 was previously described in four patients with vestibular impairment, congenital hearing loss, and inner ear abnormalities (e.g., enlarged, abnormally shaped vestibule with absent horizontal and posterior semicircular canals) [43]. This deleted 10q region contained 20 genes, including *HMX3* and *HMX2*. In another previous study, a synonymous variant *HMX3* c.114C>T was identified in two patients with superior semicircular canal dehiscence; however, no additional evidence was presented to show that this synonymous variant is potentially functional [37]. The rare deleterious missense *HMX3* variant c.534G>T (p.(Glu178Asp)) that was identified in patient 3 (Table 2) further lends support to *HMX3* as another strong candidate gene for non-syndromic vestibulopathy in humans.

*LAMA2* encodes for the alpha-2 subunit of laminin-2, is highly expressed in the brain and Schwann cells, and plays a role in neuronal migration [44]. *LAMA2* variants are associated with congenital muscular dystrophy type 1A (MDC1A), a condition characterized by severe motor weakness within the first six months of life [45]. One patient with MDC1A was reported to exhibit severe panic disorders and vertigo triggered by watching moving stimuli and rapid head movement [46]. The mouse model for human congenital muscular dystrophy (C57BL/6J-Lama2dy) had increased ABR thresholds and abnormal cochlear and vestibular structural malformations, in addition to muscular deficits in the homozygous (*dy/dy*) state [29]. These vestibular structural malformations included fibrous deposition throughout the membranous labyrinth and degeneration of the sensory structures in the semicircular crista and saccular macula. Patient 5 did not present with headaches, muscular weakness, or hearing loss and had normal inner ear structures on imaging. She did have a family history of migraine and right-sided non-syndromic vestibular weakness. It should be noted that patient 5 had variants in one migraine gene, *SLC35D2*, and two inner ear genes, *OTOP1* and *LAMA2.*

The majority of the identified variants were found in genes previously associated with migraine (*LRP1*, *ECM1*, *CARF*, *CTIF*, *RNF213*, *APOE*, *SLC35D2*), with two different rare *LRP1* variants each identified in two unrelated children with vertigo. This finding supports the hypothesis that migraine and vertigo may share some common genetic pathways. About a quarter of migraineurs report vertigo episodes either co-occurring with their headache or as separate episodes [47]. Moreover, about 30–50% of adult patients with vertigo complain of migraines [48]. Previous attempts at determining key genes in migraine-associated vertigo suggest the involvement of ion channel genes, such as *CACNA1A*, *ATP1A2*, and *SCN1A* [49]. In this study, we identified rare damaging variants in *OTOP1* and *OTOP2* which encode proton-selective ion channels (Table 2 and Appendix A). Other than these, network analyses revealed vesicular transport, membrane trafficking, coagulation or hemostasis, and the regulation of apoptosis as the most significant pathways (Figure 1, Table 3). Some of these pathways make sense in light of previous studies; for example, thrombophilic conditions were previously described in patients with migraine [50,51] and acute unilateral vestibular paresis [52]. Additionally, in an exome sequencing study of early onset Meniere’s disease patients, network analysis revealed the genes involved in cell death, with the apoptosis of hair cells being suggested as a potential pathologic mechanism [53]. However, familial Meniere’s disease is a polygenic disorder with different inheritance models, including autosomal dominant [54], autosomal recessive or compound heterozygous [55], and digenic inheritance [56]. Therefore, although there seems to be some overlap in the genes and pathways involved in pediatric vertigo, migraine, and Meniere’s disease, the observation that the children with non-syndromic vertigo have multiple rare coding variants in several genes involved in different phenotypes or pathways might indicate that the unique clinical presentation of each child with vertigo is due to the ultra-rare combination of coding variants identified in each patient.

Certain limitations of this study must be considered. First, there is difficulty in assessing vertigo from different forms of dizziness, particularly in the pediatric population. Even with the aid of advanced computer algorithms but without genetic data, diagnostic accuracy in the assessment of the dizzy patient is estimated at around 65% [57]. Second, this study used a list of candidate genes based on known inner-ear-expressed genes, genes previously associated with balance disorders in mice, and migraine genes. Other potentially novel genes may be involved with vestibulopathy in our patients; however, our approach was limited due to our sample size. Third, the variants identified in this study should be further assessed for their functional effects using variant-specific in vitro or animal model studies. However, such functional studies focusing on single genes or variants will not be able to fully replicate the effects of multiple (≥2) rare variants per patient. The ultra-rare combinations of variants that result in unique clinical presentations per patient also make it difficult to replicate findings in other probands or families. For example, through the Undiagnosed Diseases Network (UDN), we attempted to identify additional patients with undiagnosed rare diseases and also rare variants in the same candidate genes reported here but did not identify patients with a strong overlap in phenotypes, due in part to the lack of detailed audiovestibular examinations in UDN patients. Finally, only rare (MAF ≤ 0.001) coding variants were included in this study and we were not able to explore non-coding or structural variants. It is possible that the combination of rare and common variants may be responsible for vestibulopathies in pediatric patients, but studying interactions between rare and common variants for vertigo will require large-scale association studies using whole-genome data. Due to these limitations that affect the current state of genomic sciences overall, we thought it would be important to publish these candidate genes in order to facilitate rare variant identification and the genetic diagnosis of other patients with similar phenotypes and variants in the same genes. Our hope is that, eventually, the sequencing of rare variants will be useful to tailor the diagnosis and management of individuals with vestibular disorders.

## 5. Conclusions

In summary, novel and rare missense variants were identified in unrelated children with early onset vestibular dysfunction, which were supported by: the identification of additional rare variants in older patients with Meniere’s disease or LSCC asymmetry with AIS; a strong expression in mouse and human vestibular tissues; and/or previously published mouse models with vestibular deficits and structural anomalies of the vestibular apparatus. Multiple rare variants were found in each patient, suggesting that earlier vestibular dysfunction may be due to the combined effects of rare coding variants in different genes. Among the variant-containing genes, *OTOP1*, *HMX3*, and *LAMA2* are strong candidates for vestibulopathy as the mouse models for these genes demonstrated both behavioral and structural abnormalities. The follow-up of patients is recommended, particularly for children with rare damaging variants in migraine-associated genes but currently without migraine symptoms. In the future, a better understanding of the genetic basis of peripheral vestibular diseases will lead to improved diagnostics through genetic screening and tailored patient care.

## Figures and Tables

**Figure 1 genes-14-00831-f001:**
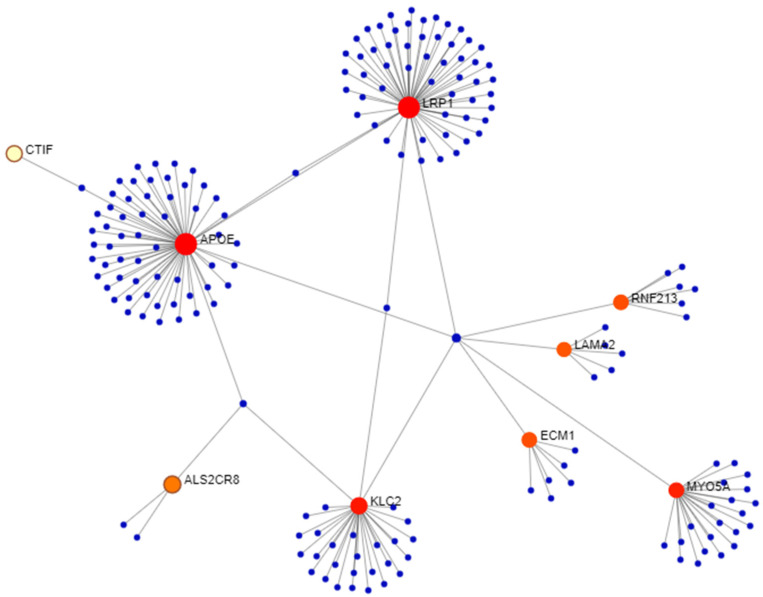
Subnetwork connecting genes with rare variants identified in pediatric patients with peripheral vestibular dysfunction. Red and yellow nodes indicate input genes (Table 2). *CARF* is included in the subnetwork though not labelled in the figure.

**Figure 2 genes-14-00831-f002:**
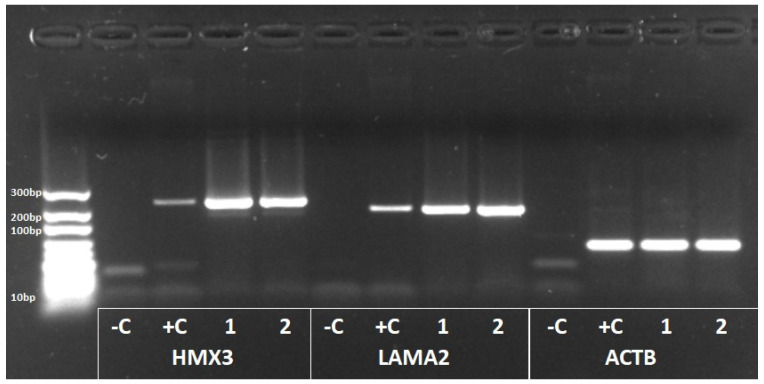
Gel image of RT-PCR products showing strong expression of *HMX3* and *LAMA2* in human vestibular tissues. (-)C: negative control; +C: positive control; 1 and 2: patients 1 and 2; *ACTB*: β-actin.

**Table 3 genes-14-00831-t003:** Top pathways per database from network analysis of candidate genes identified for pediatric vestibular disorders.

PANTHER (Biological Processes)		Gene Ontology (Biological Processes)		REACTOME	
Pathway	FDR-adj-*p*	Pathway	FDR-adj-*p*	Pathway	FDR-adj-*p*
Receptor-mediated endocytosis	2.60 × 10^–7^	Vesicle-mediated transport	1.19 × 10^–10^	Activation of BH3-only proteins	8.86 × 10^–13^
Regulation of binding	8.70 × 10^–6^	Coagulation	1.71× 10^–10^	Intrinsic pathway for apoptosis	1.33 × 10^–10^
Protein targeting	2.10 × 10^–5^	Transcription initiation from RNA polymerase II promoter	4.07 × 10^–10^	Translocation of GLUT4 to the plasma membrane	6.39 × 10^–10^
Viral process	2.20 × 10^–5^	Blood coagulation	6.91 × 10^–10^	Activation of BAD and translocation to mitochondria	3.11 × 10^–9^
Negative regulation of apoptotic process	6.30 × 10^–5^	Hemostasis	8.77 × 10^–10^	Membrane trafficking	1.11 × 10^–8^
Chemical synaptic transmission	1.30 × 10^–4^	Regulation of body fluid levels	2.59 × 10^–9^	Hemostasis	4.02 × 10^–8^
Cholesterol metabolic process	1.10 × 10^–3^	Wound healing	5.08 × 10^–9^	Platelet degranulation	9.93 × 10^–7^
Endocytosis	2.90 × 10^–3^	DNA-dependent transcription, initiation	5.30 × 10^–9^	Platelet activation, signaling, and aggregation	1.29 × 10^–6^
Heart development	4.30 × 10^–3^	Exocytosis	1.64 × 10^–8^	Insulin processing	1.32 × 10^–6^
Vitamin metabolic process	4.50 × 10^–3^	Cell morphogenesis involved in differentiation	1.71 × 10^–8^	Response to elevated platelet cytosolic Ca^2+^	1.65 × 10^–6^
		Protein localization	1.26 × 10^–6^		

Pathways that are identified in multiple databases are underlined.

**Table 4 genes-14-00831-t004:** Rare variants in Spanish probands with Meniere’s disease.

ID	Gene	Variant	Highest MAF	Damaging Prediction	Scaled CADD
S19	*ECM1*	c.844C>T (p.(Arg282Trp)	(AU)Lat: 1.3 × 10^–4^	PR, SI	24.6
60	*OTOP1*	c.380A>T (p.(His127Leu))	(AU)Eur: 7.3 × 10^–5^	PP, MT	22.8
9	*OTOP2*	c.760G>A (p.(Ala254Thr))	(AU)Afr: 2.2 × 10^–4^	PP, MT	23.6

Abbreviations: Afr: African; (AU): All of Us database; CADD: Combined Annotation-Dependent Depletion; Eur: European; Lat: Latin American; MAF: minor allele frequency; MT: MutationTaster; PP: PolyPhen2; PR: PROVEAN; SI: SIFT.

## Data Availability

Variant information is being deposited to the ClinVar database.

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
