# Peer review of "Rare Coding Variants in Patients with Non-Syndromic Vestibular Dysfunction"

_genes, 2023, doi:10.3390/genes14040831_

Round 1
Reviewer 1 Report
Sumalde et al. identified the rare coding variants in patients with non-syndromic vestibular dysfunction. Overall, this is well-written and concise.
· The author checked the allelic frequency in gnomAD v3.1.2. Did the author check the allelic frequency of the variants in the following databases, Bravo, all of us, and gnomAD v2.1.1?
· In exome analysis, did you observe any causative variant in the gene involved in age-related hearing loss?
Author Response
Sumalde et al. identified the rare coding variants in patients with non-syndromic vestibular dysfunction. Overall, this is well-written and concise.
Thank you.
- The author checked the allelic frequency in gnomAD v3.1.2. Did the author check the allelic frequency of the variants in the following databases, Bravo, all of us, and gnomAD v2.1.1?
Reply: We initially looked at MAF in the gnomAD v2.1.1 database then switched to gnomAD v3.1.2 when the data became available. All the identified variants were rare in both gnomAD versions. We now include the highest MAF values from gnomAD v3.1.2, GME, Bravo and All of Us in Table 2, Table 4, and Table S4.
- In exome analysis, did you observe any causative variant in the gene involved in age-related hearing loss?
Reply: Because Cohort 1 included only pediatric patients with vestibular disease, we did not specifically look for variants in genes involved with age-related hearing loss. It should be noted that several genes identified for age-related hearing loss in genome-wide associations studies were first identified as genes for childhood hearing loss, which we included in our candidate gene list (Table S2). However, because none of the children in cohort 1 had hearing loss, genes that were primarily involved with hearing loss but not vestibular disorders were eventually excluded (Table S3).
Reviewer 2 Report
The authors identified « Rare coding variants in patients with non-syndromic vestibular dysfunction ». This work is very interesting. However, some concerns should be made to improve the manuscript.
1- The use of a figure to describe the workflow is imperative (name of the cohort, brief clinical description, number of patients, type of analyses performed for each cohort).
2- In materials and methods section, the authors did not mention all pathogenicity prediction software (PolyPhen 2, SIFT, PROVEAN).
3- In the legend of the Table 2, two abbreviations were mentioned but they are absent in the Table (AMR and FA). What is the significance of FATHMM?
4- There is no information about the parents, except for Patient 4 for which the mother carried both MYO5 variants in cis. In materials and methods section, the authors indicated familial history of vertigo. It would be interesting to know which parent has also signs of vertigo and which variants they carry. For example, if the mother of patient 4 does not have vertigo, you can exclude this gene as responsible for vertigo in this family. It is reinforced by the case of the patient SANT116 (the cousin did not carry the MYO5 variant, but he also suffers from Meniere’s disease).
5- The power of genetics is limited, certainly due to high heterogeneity between the different cohorts. EMC1, OTOP1 and OTOP2 seem to be good candidate genes for Meniere’s disease. Did the authors consider the possibility of a de novo variant for the sporadic patient with ID=60 ?
6- What do you use as control population? Only the exomes and genomes reported in the databases?
Author Response
The authors identified « Rare coding variants in patients with non-syndromic vestibular dysfunction ». This work is very interesting. However, some concerns should be made to improve the manuscript.
- The use of a figure to describe the workflow is imperative (name of the cohort, brief clinical description, number of patients, type of analyses performed for each cohort).
Reply: A flowchart was added as Figure S1.
- In materials and methods section, the authors did not mention all pathogenicity prediction software (PolyPhen 2, SIFT, PROVEAN).
Reply: Bioinformatics tools were added to the Materials and Methods section, under 2.3 Variant prioritization.
- In the legend of the Table 2, two abbreviations were mentioned but they are absent in the Table (AMR and FA). What is the significance of FATHMM?
Reply: Thanks for noting this. The abbreviation AMR was removed from the footnote. Three out of the 17 variants listed in Table 1 were considered damaging by FATHMM.
- There is no information about the parents, except for Patient 4 for which the mother carried both MYO5 variants in cis. In materials and methods section, the authors indicated familial history of vertigo. It would be interesting to know which parent has also signs of vertigo and which variants they carry. For example, if the mother of patient 4 does not have vertigo, you can exclude this gene as responsible for vertigo in this family. It is reinforced by the case of the patient SANT116 (the cousin did not carry the MYO5 variant, but he also suffers from Meniere’s disease).
Reply: The occurrence of variants in maternal DNA was noted (Table 2) and may strengthen the potential role of a variant in a migraine, inner ear or musculoskeletal gene if there is family history of migraine, vertigo or neuromuscular disorders (Table 1). Conversely, if there is no family history of disease and the variant is inherited from the mother, this slightly weakens the case for the variant being disease-causal in the child. However, we cannot completely rule out the role of any of the variants in the child’s phenotype without complete neurologic examination of both parents and identification (or lack thereof) of the variants in paternal DNA. We tried as best as we can to obtain DNA samples from the fathers of the five children from cohort 1, but to no avail.
Regarding the MYO5A variant in the two Meniere’s disease families, the All of Us European MAF was >0.001, therefore the variant was excluded. Likewise the OTOP2 c.1609G>A (p.(Val537Ile) variant that was found in three sporadic Meniere’s disease patients was excluded due to All of Us European MAF=0.011.
5- The power of genetics is limited, certainly due to high heterogeneity between the different cohorts. EMC1, OTOP1 and OTOP2 seem to be good candidate genes for Meniere’s disease. Did the authors consider the possibility of a de novo variant for the sporadic patient with ID=60 ?
Reply: Patient ID=60 was a 77-year-old individual and we do not have DNA samples from the parents. Given the low MAF of the variant (gnomAD NFE=0.00004; All of Us Eur=0.00007) it could be a de novo mutation, but we cannot confirm it, because parent samples are not available.
6- What do you use as control population? Only the exomes and genomes reported in the databases?
Reply: We used the MAFs in the variant databases to determine if identified variants were rare. This is appropriate because the patients in our cohorts were either non-Finnish European-descent or Hispanic, and these two ethnic groups are well-represented in the databases. Additionally, we looked at variant MAF in all available ethnic groups to determine if the variants are considered rare in multiple populations and not just the population which the patient who carries the specific variant belongs to.
Round 2
Reviewer 2 Report
The authors responded to my comments. The manuscript has been sufficiently improved to warrant publication in Genes.